# Characterization of L-amino Acid Oxidase Derived from *Crotalus adamanteus* Venom: Procoagulant and Anticoagulant Activities

**DOI:** 10.3390/ijms20194853

**Published:** 2019-09-30

**Authors:** Vance G. Nielsen

**Affiliations:** Department of Anesthesiology, University of Arizona College of Medicine, Tucson, AZ 85719, USA; vgnielsen333@gmail.com

**Keywords:** procoagulant, anticoagulant, L-amino acid oxidase, thrombelastography, fibrinogen polymerizing enzyme, carbon monoxide releasing molecule

## Abstract

Snake venom enzymes of the L-amino acid oxidase (LAAO) class are responsible for tissue hemorrhage, edema, and derangement of platelet function. However, what role, if any, these flavoenzymes play in altering plasmatic coagulation have not been well defined. Using coagulation kinetomic analyses (thrombelastograph-based), it was determined that the LAAO derived from *Crotalus adamanteus* venom displayed a procoagulant activity associated with weak clot strength (no factor XIII activation) similar to thrombin-like enzymes. The procoagulant activity was not modified in the presence of reduced glutathione, demonstrating that the procoagulant activity was likely due to deamination, and not hydrogen peroxide generation by the LAAO. Further, unlike the raw venom of the same species, the purified LAAO was not inhibited by carbon monoxide releasing molecule-2 (CORM-2). Lastly, exposure of the enzyme to phenylmethylsulfonyl fluoride (PMSF) resulted in the LAAO expressing anticoagulant activity, preventing contact activation generated thrombin from forming a clot. In sum, this investigation for the first time characterized the LAAO of a snake venom as both a fibrinogen polymerizing and an anticoagulant enzyme acting via oxidative deamination and not proteolysis as is the case with thrombin-like enzymes (e.g., serine proteases). Using this thrombelastographic approach, future investigation of purified enzymes can define their biochemical nature.

## 1. Introduction

Purified, isolated enzymes derived from snake venoms have been extensively investigated to mechanistically explain toxic hypercoagulation or anticoagulation clinically [1]. While enzyme classes such as metalloproteinases, serine proteases and phospholipases have had clear mechanisms defined in disturbances in human coagulation [1], other important enzyme classes remain relatively far less studied. In particular, snake venom L-amino acid oxidases (LAAO) are responsible for tissue hemorrhage, edema and pro- or antiplatelet effects [2]. The LAAO are dimeric, glycosylated flavoenzymes that catalyze a variety of L-amino acids in the presence of oxygen to yield α-keto acids and hydrogen peroxide (H_2_O_2_) [2]. While they are not metalloproteinases, the LAAO activities are modulated by various metal ions and metal chelators. Further, while the LAAO do not have a serine in their catalytic sites, they are inhibitable with some serine protease inhibitors [3,4,5]. Specifically, the range of inhibition of various LAAO activities by ethylenediaminetetraacetic acid (EDTA) was 0–30% whereas there was much greater variation observed with phenylmethylsulfonyl fluoride (PMSF), with inhibition observed at 20–90% [3,4,5]. The purified preparations of the LAAO are very stable in solution at 4 °C, without a significant change in activity for weeks [6]. In sum, snake venom LAAO are medically important enzymes that have unusual characteristics that are distinctly different from metalloproteinases and serine proteases.

However, in contrast to snake venom metalloproteinases or serine proteinases, the LAAO mediated effects on plasmatic coagulation or fibrinolysis has remained essentially untested. One preliminary report demonstrated a selective inhibition of coagulation factor IX (FIX) by a purified LAAO obtained from *Agkistrodon halys blomhoffii* venom [7]. However, in a follow-up investigation by another group investigating the same purified protein, it was determined that the anticoagulant effects of the LAAO were secondary to a contaminant metalloproteinase [8]. Of interest, the original investigation included thrombelastography as part of their analyses [7], and the data was consistent with a contact protein inhibitor mediating anticoagulation when compared to data obtained from FIX deficient plasma [9], a pattern yet to be demonstrated with an isolated metalloproteinase. Considered as a whole, the effects, if any, of the LAAO on coagulation remained undefined to the present.

One particular LAAO that has been well studied seemed a reasonable candidate to test, and that is the enzyme purified from the Eastern diamondback rattlesnake, *Crotalus adamanteus*, which has at least three isoforms [10]. First, it is commercially available for all to assess, with stability in solution at 0–4 °C per vendor documentation (Sigma-Aldrich, St. Louis, MO, USA). Second, thrombelastography can identify the location within the coagulation pathways where LAAO may act. Third, it would be simple to expose the LAAO to potential inhibiting agents, such as carbon monoxide releasing molecules (CORM) [11,12,13]. CORM have various metal centers (e.g., boron, manganese) with carbon monoxide attached, and with ruthenium (Ru)-based CORM used in several investigations [11,12,13,14,15]. In sum, a thrombelastograph-based characterization of the LAAO coagulation kinetic behavior and vulnerability to inhibition by CORM should be facile and easily repeatable by other investigators. 

Given the aforementioned, the present investigation proposed the following goals. First, the effects (if any) of *Crotalus adamanteus* derived LAAO on human plasmatic coagulation was to be characterized with thrombelastography. Second, using either various biochemical conditions or coagulation factor deficient plasmas, this study determined which coagulation enzymes or molecules were affected by the LAAO activity to characterize its mechanism of action. Third, given that the enzyme could be susceptible to Ru-based CORM exposure via the interaction of carbon monoxide with potential heme groups [11,12,13] or with its catalytic site histidine residue [14] as has been demonstrated with other proteins [15], the experiments were planned with exposure of the LAAO to tricarbonyldichlororuthenium(II) dimer (CORM-2). Lastly, to complete its kinetic profile in response to metalloproteinase and serine protease inhibitors to compare with other LAAO profiles [3,4,5], *Crotalus adamanteus* derived LAAO was exposed to EDTA and PMSF, respectively.

## 2. Results

### 2.1. Effects of LAAO on Human Plasma Coagulation Kinetics Assessed with Thrombelastography 

The effects of the LAAO derived from *Agkistrodon halys blomhoffii* venom on human coagulation were originally described with concentrations ranging from 3.7 to 379 nM [7], so a similar range of concentrations were used to assess the LAAO activity derived from *Crotalus adamanteus* venom. The initial concentrations used (*n* = 2 per concentration) were 0, 31.25, 62.5, 125, 250 and 500 nM. This preliminary evaluation of the LAAO demonstrated a decreased time to the maximum thrombus generation (TMRTG, minutes—a measure of time to the onset of coagulation), the maximum rate of thrombus generation (MRTG, dynes/cm^2^/sec—a measure of the velocity of clot growth) and the total thrombus generation (TTG, dynes/cm^2^—a measure of clot strength) values across the range of enzyme concentrations tested. This particular pattern of deranged coagulation kinetics is similar to a thrombin-like enzyme (TLE) that directly polymerizes fibrinogen without activating factor XIII (FXIII) as has been previously described by this laboratory [12,16] and other investigators. However, as the molecular action of the LAAO (oxidative deamination) is unlike TLE (proteolytic, cleaving α and β chains of fibrinogen), the label fibrinogen polymerizing enzyme (FPE) is used to describe the action of this enzyme. This kinetic behavior would classify this enzyme as a weak procoagulant, and not an anticoagulant as reported with the previous LAAO derived from *Agkistrodon halys blomhoffii* venom [7]. The corresponding traditional thrombelastographic signature with the corresponding parametric values (TMRTG, MRTG and TTG) and the differences between the control clot formation to that of plasma exposed to 500 nM LAAO are displayed in Figure 1A,B.

For better characterization of the kinetic behavior of this LAAO, an additional set of experiments incorporating a range of 0, 5, 50, 500 nM (*n* = 6 replicates per concentration) were performed to facilitate the statistical analysis. The results are displayed in Figure 1C. As the concentration of the LAAO increased, the TMRTG values became significantly smaller. The changes in MRTG values were biphasic, with an initial decrease with the smaller concentrations of the LAAO followed by an increase in MRTG, with the greatest concentration of the LAAO. In contrast, there was a progressive decrease in TTG values with increasing LAAO values that was stable between the 50 and 500 nM concentrations. This pattern of decreasing TMRTG, biphasic MRTG and decreasing TTG is consistent with a progressive LAAO effect preceding and dominating the endogenous contact protein generated thrombin effect that occurs between the plasma and cup/pin of the thrombelastographic cup. 

Of interest, the raw venom of *Crotalus adamanteus* displays the same kinetic pattern under the same conditions [12], presumably via the action of a serine protease, not an LAAO. Another possibility, which mechanistically explains the action of the present LAAO being tested was that it was generating thrombin from the plasma while simultaneously inhibiting FXIII. Thus, additional characterization was undertaken in the next series of experiments. Given the relatively decreased TMRTG values observed with 500 nM of the LAAO, this concentration was used in all subsequently described experimentation. 

### 2.2. Effects of Heparin, Calcium and Reduced Glutathione (GSH) on LAAO Procoagulant Activity as Assessed with Thrombelastography

Given the weak, procoagulant, FPE nature of the LAAO, a series of experiments were conducted to determine the site and mechanism of action of the enzyme. First, to exclude the LAAO mediated native thrombin generation as a mechanism of action, plasma was pretreated with heparin 10 U/mL or had no addition of calcium to overcome the sodium citrate anticoagulation prior to addition of the LAAO. Further, as H_2_O_2_ derived from the LAAO could modify fibrinogen [17,18], reduced glutathione (GSH) was added to plasma to scavenge H_2_O_2_ via the action of endogenous plasmatic hydrogen peroxidase [19,20]. It has already been demonstrated that GSH at the indicated concentration has no effect on plasmatic coagulation kinetics [21]. The results are displayed in Figure 2. 

The weak procoagulant activity of the LAAO was not significantly affected by any of the plasmatic additions tested, demonstrating that this LAAO is a FPE, acting on fibrinogen directly via the modification of L-amino acids, and not modifying fibrinogen via the generation of H_2_O_2_. Lastly, *n* = 6 replicates per condition of plasma with the heparin addition or lacking calcium addition were performed. No detectable coagulation was detected after 30 min, demonstrating that the LAAO activity that was observed in Figure 1 was not secondary to the inadequate prevention of thrombin action (heparin) or generation (calcium).

### 2.3. Effects of CORM-2 on LAAO Procoagulant Activity as Assessed with Thrombelastography

As both raw venom and phospholipase A_2_ activity derived from the venom of *Crotalus adamanteus* is inhibited by CORM-2 [16,17,18], it is of interest to determine if the LAAO are vulnerable to such a Ru-based CORM as a possible future therapeutic. The results of these experiments are displayed in Figure 3.

The LAAO activity was not affected over a 10-fold range of CORM-2 concentrations, demonstrating that the enzyme was not vulnerable to carbon monoxide or reactive Ru species. As no effect was observed with both concentrations of CORM-2, no purpose would have been served by additional exposure to the inactivated carrier molecule so those experiments were not performed.

### 2.4. Effects of EDTA and PMSF on LAAO Procoagulant Activity as Assessed with Thrombelastography

To compare the effects of metalloproteinase and serine protease inhibitors on the presently investigated the LAAO with the inhibitory pattern seen with other LAAO [3,4,5], the enzyme was exposed to EDTA (0, 1 and 5 mM) or PMSF (0 and 1 mM) for 30 min at 37 °C. A 1% addition of these enzyme mixtures to plasma followed by thrombelastographic analysis was performed for 20 min as mentioned in Materials and Methods. An additional set of plasma samples had a 1% addition of 1 mM PMSF as a control condition (final concentration 10 µM) in the series of PMSF experiments. The results of the experiments with EDTA and PMSF exposure are displayed in Figure 4 and Figure 5, respectively.

As seen in Figure 4, the LAAO procoagulant activity was unaffected by EDTA exposure. Further, the LAAO activity was not dependent on calcium (Figure 2) and the final concentration of EDTA following the 1% addition of venom mixture was small (10–50 µM) and unlikely to affect coagulation in the presence of an 11 mM addition of calcium (see results). Therefore, it was not anticipated that added EDTA would interfere with LAAO or normal endogenous plasma enzyme activity. In sharp contrast as displayed in Figure 5, exposure of the LAAO to PMSF did not result in the inhibition of procoagulant activity. Rather, the LAAO displayed anticoagulant activity with no coagulation observed. To verify that this inhibition of coagulation was not secondary to the small concentration of PMSF in the final plasma mixture (10 µM), the displayed control condition was generated that was not significantly different from plasma without the PMSF addition. Given the data displayed in Figure 4 and Figure 5, there was no contaminant metalloproteinase or serine protease activity present. Lastly, these data demonstrate that depending on the additives present, the LAAO may display procoagulant or anticoagulant activity.

## 3. Discussion

This investigation achieved its stated goals. For the first time, the coagulation kinetic effects of a purified snake venom LAAO was characterized with kinetomics, and with the enzyme displaying FPE procoagulation and PMSF associated anticoagulation. The experiments with the heparin addition or calcium omission demonstrated that the LAAO must be polymerizing fibrinogen independent of endogenous thrombin generation and FXIII crosslinking. Furthermore, it was the direct deaminating action of LAAO, not the coincident generation of H_2_O_2_, that was responsible for the coagulation procoagulant profile as demonstrated by the GSH addition experiments. Further, the experiments involving CORM-2 demonstrated functionally, that the procoagulant effects of LAAO were not affected as was the phospholipase A_2_ and raw venom activity of *Crotalus adamanteus* [6,7,8]. The lack of inhibition of the LAAO activity by EDTA and the transformation of procoagulant to anticoagulant activity by PMSF further confirmed that the phenomena observed in this system were LAAO mediated. In summary, these data demonstrate that the LAAO purified from *Crotalus adamanteus* venom is a hereto unidentified FPE or anticoagulant agent acting via oxidative deamination, a mechanism that is distinct from the catalytic activity of proteolytic serine proteases and other trypsin-like enzymes [22]. 

The contribution of the LAAO to hemostatic derangements following envenomation may be a hereto an unappreciated phenomenon, and could be responsible for clinically relevant coagulopathy. While not designed to assess the impact of multiple enzymes, the experimental design and the results of the assessment of raw *Crotalus adamanteus* venom on human plasma demonstrated a significant reduction of TLE activity following exposure to CORM-2, but a hint of residual TLE or FPE activity remained [12]. However, the data collection time period was only 15 min [12], and not enough time to assess if endogenous coagulation had returned to normal after the venom was exposed to CORM-2 which typically requires 20–30 min [13]. Thus, the possibility that residual, detectable CORM-2 insensitive FPE activity remained, is an issue that deserves visitation in future investigations. In sum, while the LAAO represent only 5% of venom gland toxin transcripts [23], it may still significantly affect coagulation, contributing to the defibrinogenating of the envenomed.

The use of EDTA and PMSF to confirm that a newly isolated, sequenced protein is a metalloproteinase or serine proteases, respectively, is a classic paradigm that has been used for decades. However, as has been demonstrated numerous times, the isozymes of any given the LAAO may have activity inhibited by EDTA (0–30%) or PMSF (20–90%) [3,4,5]. The concentrations of EDTA and PMSF used in the present investigation are within the range of these previous works [3,4,5], but the systems used to assess the LAAO activity are very different from these studies. The previous works used solutions of isolated L-amino acids (L-Alanine, L-Aspartate, L-Glycine, L-Valine, etc.) to assess changes in specific LAAO activities [3,6,10], whereas this investigation assessed functional changes in coagulation. The EDTA experiments of the present work demonstrated no change in procoagulant activity by the LAAO, similar to the small amount of a decrease in activity in other LAAO [3,4,5]. However, the experiments with PMSF provided remarkable data, wherein procoagulant activity changed to anticoagulant activity displayed by the LAAO of *Crotalus adamanteus* venom. PMSF may have inhibited a procoagulant isoform of the LAAO of *Crotalus adamanteus* venom, allowing an anticoagulant isoform to be expressed, or PMSF could have in some other way modified the procoagulant isoform(s) to act as an anticoagulant. The reason why PMSF converts LAAO activity to being anticoagulant in nature is unknown, with possibilities including a change in its oxidative deamination substrate specificity, a change in amino acid target on fibrinogen, or perhaps a change in the coagulation protein target to include critical clotting factors, such as thrombin or any of the other contact protein pathway serine proteases. The determination of the precise molecular mechanism responsible for the anticoagulant activity of this LAAO is beyond the scope of the present work. In summary, the results of the experiments, wherein the LAAO purified from *Crotalus adamanteus* venom was exposed to EDTA and PMSF, demonstrate a degree of responsiveness similar to that observed with other snake venom LAAO.

The variation in inhibition observed with LAAO with PMSF is worthy of further comment, and not just for the LAAO but for other enzymes that are not serine proteases. First, PMSF does not inhibit all serine proteases [24,25], and not all enzymes that are inhibited by PMSF are serine proteases, which include LAAO [3,4,5], CoA-independent transacylase [26], diacylglycerol lipase [27], calcium-dependent cysteine proteinase [28] and the factor X activator losac, a hemolin [29]. In summary, while the interaction of PMSF with a serine residue within the enzyme catalytic site is the classic paradigm of inhibition, it is entirely possible that there may be proteins with serine residues outside the catalytic site that could modulate enzymatic activity (e.g., changes in structural conformation). 

In conclusion, this investigation characterized in human plasma the weak procoagulant nature and PMSF associated anticoagulant activity of the LAAO derived from *Crotalus adamanteus* venom. A methodical, kinetomic methodology provided preliminary insights into the functional mechanism by which procoagulation was affected. Further, the exposure to CORM-2, an agent known to inhibit other enzymes contained in the raw venom, and the exposure to EDTA and PMSF resulted in the data demonstrating that the LAAO was most likely pure and the experimental data generated by it was specific to the activity of the enzyme and not some contaminant. Using this approach, it is anticipated that future investigation of similar purified enzymes will define the biochemical nature and antagonist vulnerabilities to improve our understanding of the important contributors of the morbidity associated with envenomation. 

## 4. Materials and Methods 

### 4.1. Chemicals and Human Plasma

The purified, lyophilized LAAO isolated from *Crotalus adamanteus* venom, calcium-free phosphate buffered saline (PBS), dimethyl sulfoxide (DMSO), reduced glutathione (GSH), tricarbonyldichlororuthenium (II) dimer (CORM-2), EDTA and PMSF were obtained from Millipore Sigma (Saint Louis, MO, USA). The unfractionated heparin (1000 U/mL) was obtained from SAGENT Pharmaceuticals (Schaumburg, IL, USA). Calcium chloride (200 mM) was obtained from Haemonetics Inc. (Braintree, MA, USA). The pooled normal human plasma that was sodium citrate anticoagulated and maintained at −80 °C was obtained from George King Bio-Medical (Overland Park, KS, USA). This plasma is a commercial product obtained from consented, anonymous and compensated healthy donors by the vendor, so no further consent is needed to be obtained by end users. Lastly, LAAO was dissolved in PBS at a concentration of 6.5 mg/mL freshly for each set of experiments and kept at 4 °C.

### 4.2. Thrombelastographic Analyses

The volumes of subsequently described plasmatic and other additives summed to a final volume of 360 µL. The samples were composed of 320 µL of plasma; 16.4 µL of PBS, 20 µL of 200 mM CaCl_2_, and 3.6 µL of PBS or the LAAO solution mixture, which were pipetted into a disposable cup in a thrombelastograph^®^ hemostasis system (Model 5000, Haemonetics Inc., Braintree, MA, USA) at 37 °C, and then rapidly mixed by moving the cup up against and then away from the plastic pin five times. The LAAO was always the last constituent added prior to mixing and data collection. The following viscoelastic parameters described previously [11,12,13] were measured: time to maximum rate of thrombus generation (TMRTG)—this is the time interval (minutes) observed prior to maximum speed of clot growth; maximum rate of thrombus generation (MRTG)—this is the maximum velocity of clot growth observed (dynes/cm^2^/second); and the total thrombus generation (TTG, dynes/cm^2^)—the final viscoelastic resistance observed after clot formation. The data were collected for 20 min.

The initial concentration-response relationship of the LAAO was generated over a 0, 31.25, 62.5, 125, 250 and 500 nM range (*n* = 2 per condition). For statistical comparison, an additional set of experiments incorporating a range of 0, 5, 50, 500 nM (*n* = 6 replicates per concentration) were performed. Thereafter, a concentration of 500 nM LAAO was used in all subsequent experimentation. 

### 4.3. Heparin Addition, Calcium Omission, and Reduced Glutathione (GSH) Addition Experiments

These experiments had the same volumes of reactants with the indicated concentration of the LAAO with the exception that the three plasma condition types had the following additions. In the experiments involving heparin addition, 10 U/mL final concentration was achieved by adding 10 µL of a 1000 U/mL stock solution to each ml of plasma used. In the experiments involving calcium omission, 20 µL of PBS was added to the plasma mix instead of 20 µL of calcium chloride. Lastly, 50 µL of 20 mM stock GSH was added to each ml of plasma to obtain a final concentration of 1 mM. 

### 4.4. CORM-2 Addition Experiments

In the experiments with CORM-2, the conditions utilized were: (1) control condition—no LAAO, DMSO 1% addition (*v/v*) in PBS; (2) LAAO condition—LAAO (500 nM final concentration in plasma), DMSO 1% addition (*v/v*) in PBS; (3) C1 condition—LAAO, CORM-2 1% addition in DMSO (100 µM final concentration); (4) C2 condition—LAAO, CORM-2 1% addition in DMSO (1 mM final concentration). The solutions were incubated for 5 min at room temperature, and then 3.6 µL of one of these solutions was added to the plasma sample in the plastic cup.

### 4.5. EDTA and PMSF Addition Experiments

In the experiments with EDTA, the LAAO was exposed to 0, 1, or 5 mM EDTA for 30 min at 37 °C. Thereafter, a 1% addition of these solutions were made to a plasma sample mixture as outlined in Section 4.2 and an analysis conducted for 20 min. The conditions for the PMSF experiments included the LAAO exposed to 0 or 1 mM PMSF in PBS or just PBS with 1 mM PMSF for 30 min at 37 °C. Thereafter, 3.6 µL (1% addition, *v/v*) of one these solutions were made to a plasma sample mixture and the data was collected for 20 min. The final concentration of PMSF following this 1% addition of a 1 mM solution to plasma samples was10 µM.

### 4.6. Statistical Analyses

The data are presented as the mean ± SD. The graphics were generated with a commercially available program (Origen2019b, OrigenLab Corporation, Northampton, MA, USA). The experimental conditions were composed of *n* = 6 replicates per condition as this provides a statistical power > 0.8 with *p* < 0.05 utilizing these techniques [11,12,13]. A statistical program was used for one-way analyses of variance (ANOVA) comparisons between the conditions, followed by Holm-Sidak post hoc analysis or unpaired, or two-tailed Student’s *t*-tests as appropriate (SigmaPlot 14, Systat Software, Inc., San Jose, CA, USA). *p* < 0.05 was considered significant. 

## Figures and Tables

**Figure 1 ijms-20-04853-f001:**
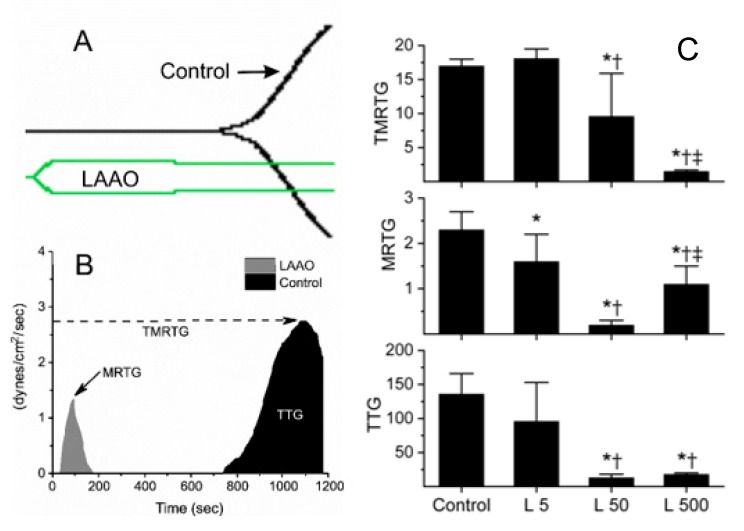
A comparison of traditional (panel **A**) and parametric (panel **B**) outputs with a control plasma sample (black trace) and a sample with 500 nM L-amino acid oxidase (LAAO) addition (green and gray traces) are provided. The concentration-response of LAAO on coagulation (panel **C**) is also displayed. The run time was 20 min for all outputs, and for clarity, the axes of panel A were omitted. Panels **B** and **C**: TMRTG (dashed line, arrow) = minutes; MRTG (solid line, arrow) = dynes/cm^2^/sec; TTG (white inset) = dynes/cm^2^. Panel C data is displayed as mean ± SD. Control = 0 nM LAAO; L5 = 5 nM LAAO; L50 = 50 nM LAAO; L500 = 500 nM LAAO. * *p* < 0.05 versus Control; ^†^
*p* < 0.05 versus L5; ^‡^
*p* < 0.05 versus L50.

**Figure 2 ijms-20-04853-f002:**
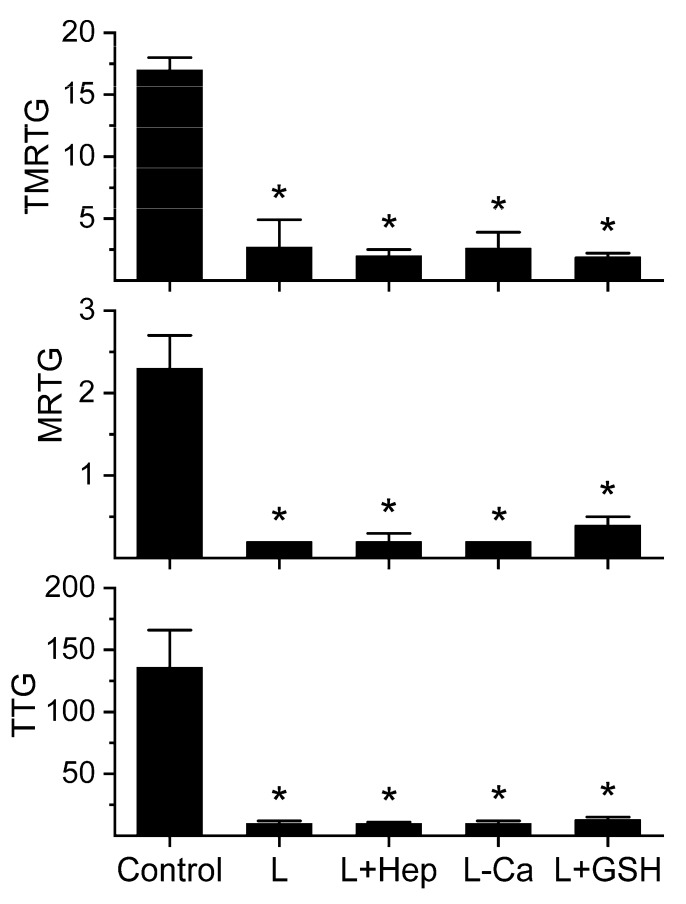
The kinetomic effects of LAAO on plasma containing heparin, no calcium, or GSH. The data are displayed as the mean ± SD. TMRTG = minutes; MRTG = dynes/cm^2^/sec; TTG = dynes/cm^2^. Control = plasma without the LAAO exposure; L = plasma with additive naive 500 nM LAAO; L + Hep = LAAO added to plasma with heparin 10 U/mL addition; L-Ca = LAAO added to plasma without calcium addition; L + GSH = LAAO added to plasma with 1 mM GSH addition. * *p* < 0.05 versus Control.

**Figure 3 ijms-20-04853-f003:**
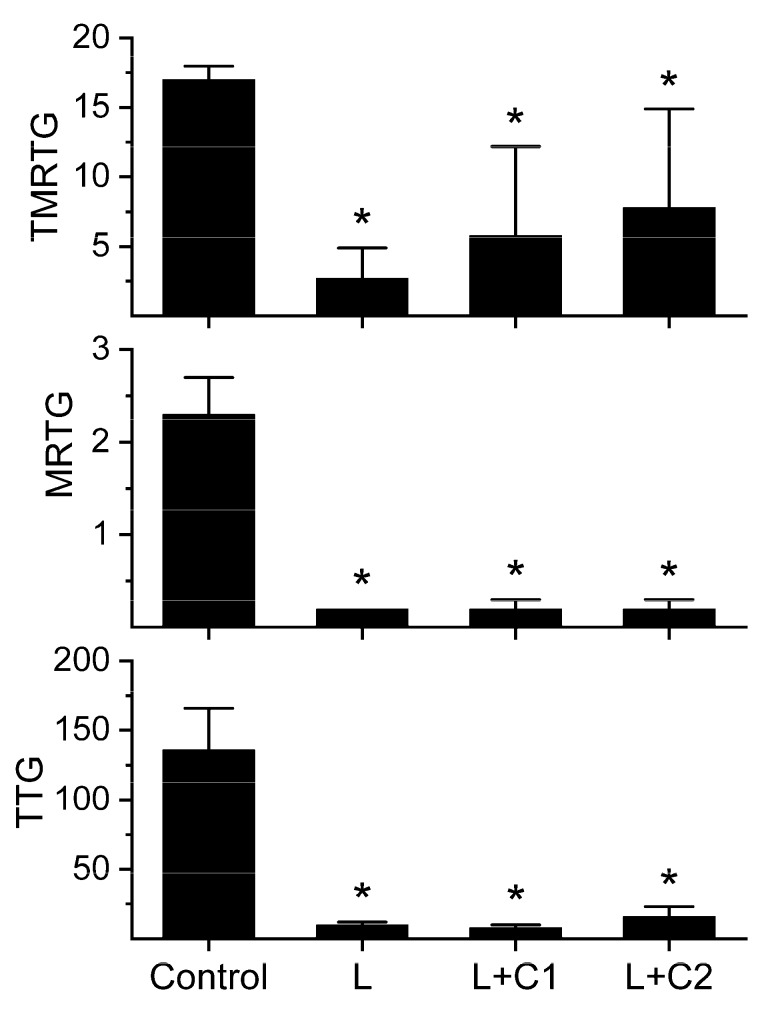
The kinetomic effects of exposure of the LAAO to CORM-2. The data are displayed as the mean ± SD. TMRTG = minutes; MRTG = dynes/cm^2^/sec; TTG = dynes/cm^2^. Control = plasma without LAAO addition; L = plasma with additive naive 500 nM LAAO addition; L + C1 = LAAO exposed to 100 µM CORM-2 prior to addition to plasma; L + C2 = LAAO exposed to 1 mM CORM-2 prior to addition to plasma. * *p* < 0.05 versus Control.

**Figure 4 ijms-20-04853-f004:**
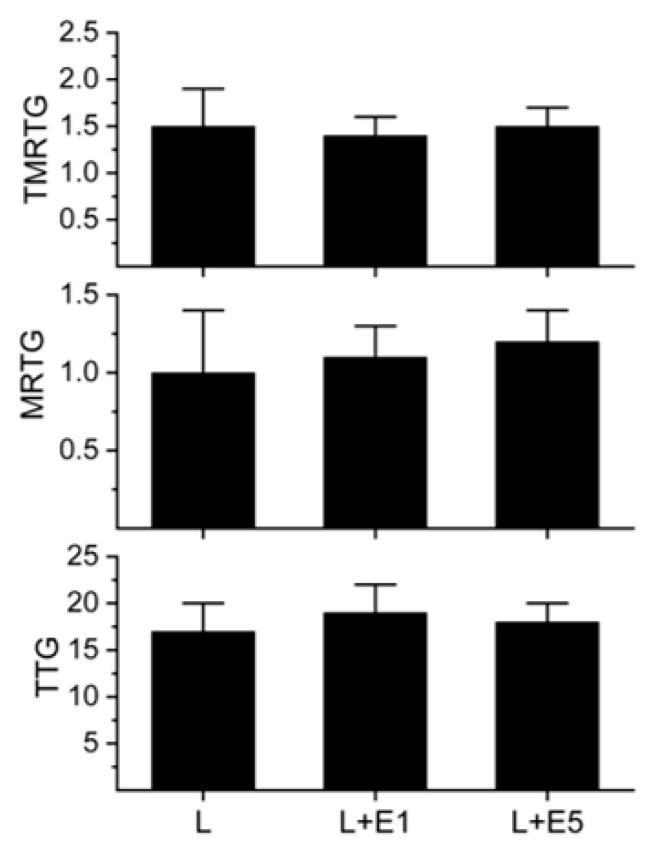
The effects of exposure of the LAAO to ethylenediaminetetraacetic acid (EDTA). The data are displayed as the mean ± SD. TMRTG = minutes; MRTG = dynes/cm^2^/sec; TTG = dynes/cm^2^. L = plasma with 500 nM LAAO addition; L + E1 = LAAO exposed to 1 mM EDTA; L + E5 = LAAO exposed to 5 mM EDTA.

**Figure 5 ijms-20-04853-f005:**
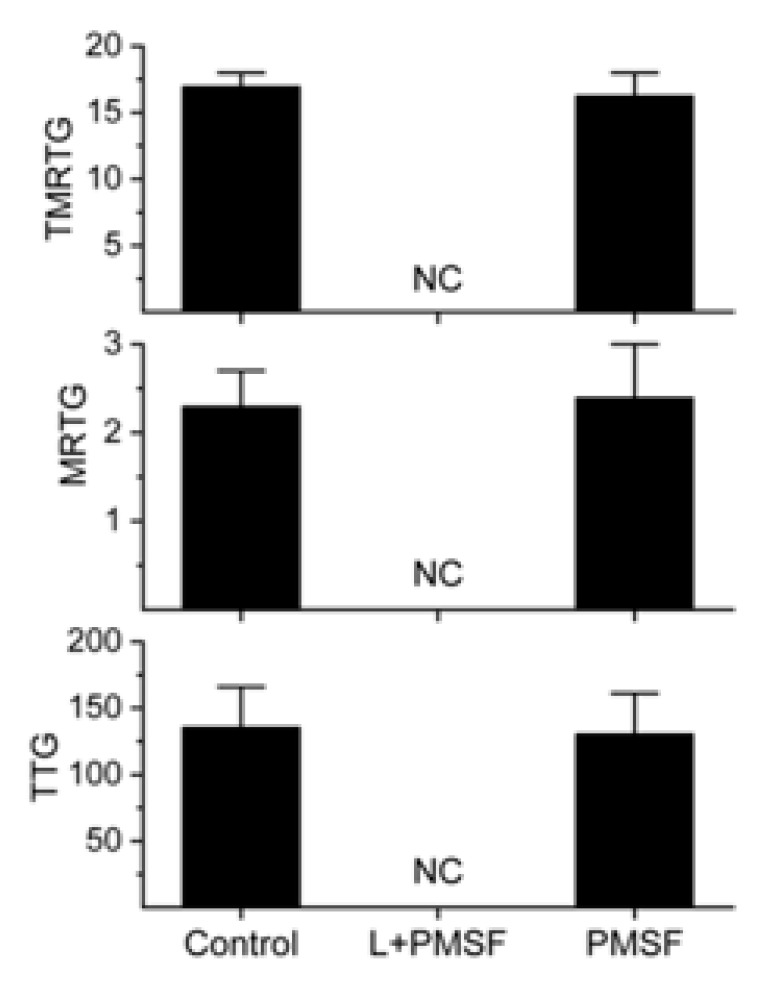
The effects of exposure of the LAAO to phenylmethylsulfonyl fluoride (PMSF). The data are displayed as the mean ± SD. TMRTG = minutes; MRTG = dynes/cm^2^/sec; TTG = dynes/cm^2^. Control = Control plasma without LAAO addition; L + PMSF = LAAO exposed to 1 mM PMSF; PMSF = plasma exposed to 10 µM PMSF; NC = no measurable coagulation.

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
