# Peer review of "Characterization of L-amino Acid Oxidase Derived from Crotalus adamanteus Venom: Procoagulant and Anticoagulant Activities"

_ijms, 2019, doi:10.3390/ijms20194853_

Round 1

Reviewer 1 Report

A brief summary

This manuscript characterizes the procoagulant and anticoagulant activities of L-amino acid oxidase derived from Crotalus adamanteus venom using thrombelastography (a coagulation kinetomic analyses procedure).   L-amino acid oxidase derived from Crotalus adamanteus venom demonstrate varying procoagulant and anticoagulant activities different from other procoagulants or anticoagulants (such as serine proteases).

Broad comments.

The manuscript is in general well written and describes the rational for using various controls and mixtures of reactants, and for further study of this class of enzymes.

Specific comments.

Some issues may obscure the meaning of the paper or readability:

Inconsistent definition of abbreviations immediately following first mention of a phrase, or an abbreviation without a previous phrase.

Line 19 (add “CORM-2”; add definition prior to “PMSF”)

Lines 48, 53 (definition line 48; abbreviation line 53 “FIX”?)

Line 121 (previous definition of FXIII?)

Line 142 (previous definition of FPE?)

Line 209 (previous definition of TLE?)

Missing words or incorrect words

Line 61 “can to”

Line 62 “to exposure”

May be some confusion of the use and meaning of “CORM” and “CORM-2” abbreviations within the text (e.g. lines 72, 74) and table of abbreviations. What are the relationships (chemical, structural, etc.) between CORM and CORM-2?

Line 318. Abbreviations table may need to include any abbreviations used in the text, including clotting factors.  It may be helpful to a reader (i.e. student) to describe PMSF as a serine protease inhibitor.

Author Response

“A brief summary”

“This manuscript characterizes the procoagulant and anticoagulant activities of L-amino acid oxidase derived from Crotalus adamanteus venom using thrombelastography (a coagulation kinetomic analyses procedure).   L-amino acid oxidase derived from Crotalus adamanteus venom demonstrate varying procoagulant and anticoagulant activities different from other procoagulants or anticoagulants (such as serine proteases).”

“Broad comments.”

“The manuscript is in general well written and describes the rational for using various controls and mixtures of reactants, and for further study of this class of enzymes.”

“Specific comments.”

“Some issues may obscure the meaning of the paper or readability:”

“Inconsistent definition of abbreviations immediately following first mention of a phrase, or an abbreviation without a previous phrase.”

“Line 19 (add “CORM-2”; add definition prior to “PMSF”)”  Thank you for this suggestion, which was implemented.

“Lines 48, 53 (definition line 48; abbreviation line 53 “FIX”?)”  Thank you for catching this; I have placed the abbreviation next to the definition on line 48.

“Line 121 (previous definition of FXIII?)”  Yes, there was a previous definition on lines 89-90.

“Line 142 (previous definition of FPE?)”  Yes, there was a previous definition in line 92.

“Line 209 (previous definition of TLE?)”  Yes, there was a previous definition in line 89.

“Missing words or incorrect words”

“Line 61 “can to””  Thank you; I have deleted “to” to correct the sentence.

“Line 62 “to exposure””  Thank you; I have changed “exposure” to “expose”.

“May be some confusion of the use and meaning of “CORM” and “CORM-2” abbreviations within the text (e.g. lines 72, 74) and table of abbreviations. What are the relationships (chemical, structural, etc.) between CORM and CORM-2?”  Thank you for this comment.  I have added a sentence across lines 63-65 to note that CORM can have various metal centers, and Ru-based CORM such as CORM-2 have been used in investigations.  I have also modified the Abbreviations to indicate differences between CORM and CORM-2 for clarity.

“Line 318. Abbreviations table may need to include any abbreviations used in the text, including clotting factors.  It may be helpful to a reader (i.e. student) to describe PMSF as a serine protease inhibitor.”  I have made the requested change in Abbreviations.

“A brief summary”

“This manuscript characterizes the procoagulant and anticoagulant activities of L-amino acid oxidase derived from Crotalus adamanteus venom using thrombelastography (a coagulation kinetomic analyses procedure).   L-amino acid oxidase derived from Crotalus adamanteus venom demonstrate varying procoagulant and anticoagulant activities different from other procoagulants or anticoagulants (such as serine proteases).”

“Broad comments.”

“The manuscript is in general well written and describes the rational for using various controls and mixtures of reactants, and for further study of this class of enzymes.”

“Specific comments.”

“Some issues may obscure the meaning of the paper or readability:”

“Inconsistent definition of abbreviations immediately following first mention of a phrase, or an abbreviation without a previous phrase.”

“Line 19 (add “CORM-2”; add definition prior to “PMSF”)”  Thank you for this suggestion, which was implemented.

“Lines 48, 53 (definition line 48; abbreviation line 53 “FIX”?)”  Thank you for catching this; I have placed the abbreviation next to the definition on line 48.

“Line 121 (previous definition of FXIII?)”  Yes, there was a previous definition on lines 89-90.

“Line 142 (previous definition of FPE?)”  Yes, there was a previous definition in line 92.

“Line 209 (previous definition of TLE?)”  Yes, there was a previous definition in line 89.

“Missing words or incorrect words”

“Line 61 “can to””  Thank you; I have deleted “to” to correct the sentence.

“Line 62 “to exposure””  Thank you; I have changed “exposure” to “expose”.

“May be some confusion of the use and meaning of “CORM” and “CORM-2” abbreviations within the text (e.g. lines 72, 74) and table of abbreviations. What are the relationships (chemical, structural, etc.) between CORM and CORM-2?”  Thank you for this comment.  I have added a sentence across lines 63-65 to note that CORM can have various metal centers, and Ru-based CORM such as CORM-2 have been used in investigations.  I have also modified the Abbreviations to indicate differences between CORM and CORM-2 for clarity.

“Line 318. Abbreviations table may need to include any abbreviations used in the text, including clotting factors.  It may be helpful to a reader (i.e. student) to describe PMSF as a serine protease inhibitor.”  I have made the requested change in Abbreviations.

“A brief summary”

“This manuscript characterizes the procoagulant and anticoagulant activities of L-amino acid oxidase derived from Crotalus adamanteus venom using thrombelastography (a coagulation kinetomic analyses procedure).   L-amino acid oxidase derived from Crotalus adamanteus venom demonstrate varying procoagulant and anticoagulant activities different from other procoagulants or anticoagulants (such as serine proteases).”

“Broad comments.”

“The manuscript is in general well written and describes the rational for using various controls and mixtures of reactants, and for further study of this class of enzymes.”

“Specific comments.”

“Some issues may obscure the meaning of the paper or readability:”

“Inconsistent definition of abbreviations immediately following first mention of a phrase, or an abbreviation without a previous phrase.”

“Line 19 (add “CORM-2”; add definition prior to “PMSF”)”  Thank you for this suggestion, which was implemented.

“Lines 48, 53 (definition line 48; abbreviation line 53 “FIX”?)”  Thank you for catching this; I have placed the abbreviation next to the definition on line 48.

“Line 121 (previous definition of FXIII?)”  Yes, there was a previous definition on lines 89-90.

“Line 142 (previous definition of FPE?)”  Yes, there was a previous definition in line 92.

“Line 209 (previous definition of TLE?)”  Yes, there was a previous definition in line 89.

“Missing words or incorrect words”

“Line 61 “can to””  Thank you; I have deleted “to” to correct the sentence.

“Line 62 “to exposure””  Thank you; I have changed “exposure” to “expose”.

“May be some confusion of the use and meaning of “CORM” and “CORM-2” abbreviations within the text (e.g. lines 72, 74) and table of abbreviations. What are the relationships (chemical, structural, etc.) between CORM and CORM-2?”  Thank you for this comment.  I have added a sentence across lines 63-65 to note that CORM can have various metal centers, and Ru-based CORM such as CORM-2 have been used in investigations.  I have also modified the Abbreviations to indicate differences between CORM and CORM-2 for clarity.

“Line 318. Abbreviations table may need to include any abbreviations used in the text, including clotting factors.  It may be helpful to a reader (i.e. student) to describe PMSF as a serine protease inhibitor.”  I have made the requested change in Abbreviations.

Reviewer 2 Report

Nielsen characterized procoagulant property of L-amino acid oxidase (LAAO) derived from Crotalus adamanteus using thrombelastographic approach. The present study may give us a new knowledge how snake venom derived LAAO cause coagulation. Although all the reagents that the author used did not affect thrombelastography induced by LAAO, the author added some interpretation of the data that sounds reasonable.

I have two questions/critics that I would like the author to answer.

1/ Figure 2 MRTG graph

There is a biphasic pattern by titration of LAAO. Could the author discuss why it happens?

2/ Figure 5&6

EDTA chelate Calcium that are needed for coagulation. However, adding EDTA did not affect thrombelastography. Similarly, PMSF inhibits serine proteases (coagulation factors). However, adding PMSF did not affect thrombelastography. Why are they?

Also, I am wondering adding EDTA and PMSF are good ways to inhibit metalloproteases and serine proteases because they affect coagulation factors that are essential for thrombelastography. Please make comment on it.

Author Response

“Nielsen characterized procoagulant property of L-amino acid oxidase (LAAO) derived from Crotalus adamanteus using thrombelastographic approach. The present study may give us a new knowledge how snake venom derived LAAO cause coagulation. Although all the reagents that the author used did not affect thrombelastography induced by LAAO, the author added some interpretation of the data that sounds reasonable.”

“I have two questions/critics that I would like the author to answer.”

“1/ Figure 2 MRTG graph”

“There is a biphasic pattern by titration of LAAO. Could the author discuss why it happens?”  Lines 115-118 had an explanation for the pattern changes in TMRTG, biphasic MRTG and TTG with increasing LAAO concentrations.  In brief, as the action of endogenous thrombin generation is outpaced by LAAO, the clotting begins sooner, with decreased speed of clot growth, followed by eventual increased maximum LAAO mediated clot growth.  This is now emphasized in this section of Results.

“2/ Figure 5&6”

“EDTA chelate Calcium that are needed for coagulation. However, adding EDTA did not affect thrombelastography. Similarly, PMSF inhibits serine proteases (coagulation factors). However, adding PMSF did not affect thrombelastography. Why are they?”  I now include my response to this question in lines 179-186.  In brief, the concentrations of EDTA are too small in the final plasma mixture to every be of concern following the three order of magnitude larger calcium addition to overcome citrate anticoagulation.  Similarly, using the indicated control in figure 6, I demonstrated experimentally that the final concentration of PMSF was insufficient to inhibit the serine protease coagulation factors in human plasma.

“Also, I am wondering adding EDTA and PMSF are good ways to inhibit metalloproteases and serine proteases because they affect coagulation factors that are essential for thrombelastography. Please make comment on it.”  Again, thank you for this concern.  When I designed these experiments, I made sure that the reactants that could by themselves could affect plasmatic coagulation were at final concentrations that would not confound my data.  This is indicated in lines 179-186.

Reviewer 3 Report

The work deals with the characterization of procoagulation and anticoagulation activities of L-amino acid oxidase derived from Crotalus adamanteus snake venom. Firstly, the effects of L-amino acid oxidase on human plasma coagulation kinetics was investigated with a three different concentration of  L-amino acid oxidase. Further, the effects of heparin, calcium, reduced glutathione, tricarbonyldichlororuthenium(II) dimer on L-amino acid oxidase procoagulant activity were determined. The paper is well structured and procedures are well described, supported by results.

However, only one minor correction might be needed, before publishing, as follows:

Line 95-100: X and Y axes for the figure 1(a) is missing. Also the sentence between line 95-100 is divided to few parts due to the mismatch of figure 1 and figure 2. It might be better to combine both figures in one file.

Author Response

“The work deals with the characterization of procoagulation and anticoagulation activities of L-amino acid oxidase derived from Crotalus adamanteus snake venom. Firstly, the effects of L-amino acid oxidase on human plasma coagulation kinetics was investigated with a three different concentration of  L-amino acid oxidase. Further, the effects of heparin, calcium, reduced glutathione, tricarbonyldichlororuthenium(II) dimer on L-amino acid oxidase procoagulant activity were determined. The paper is well structured and procedures are well described, supported by results.”

“However, only one minor correction might be needed, before publishing, as follows:”

“Line 95-100: X and Y axes for the figure 1(a) is missing. Also the sentence between line 95-100 is divided to few parts due to the mismatch of figure 1 and figure 2. It might be better to combine both figures in one file.”  I thank the reviewer for the kind comments.  In the original file submitted, the problem with text line mismatch was not present and must have occurred after processing prior to submitting it for review.  With regard to axes for panel A, the output from the thrombelastographic program that I imported into my graphics program does not have units depicted.  If I choose the overlap option of the two traces only a Y axis without X is generated.  That is why I tried to align panel A over panel B and mention that the run time was 20 minutes.  I have added a statement in the figure legend and realigned the figures to address these comments.  I hope the reviewer finds this acceptable.   

Round 2

Reviewer 2 Report

The author appropriately answered my questions.

I do not have further concerns.